# Influence of geographic access and socioeconomic characteristics on breast cancer outcomes: A systematic review

Benoit Conti[1]*, Audrey Bochaton[2], Hélène Charreire[3,4], Hélène Kitzis-Bonsang[5], Caroline Desprès[6], Sandrine Baffert[7], Charlotte Ngô[5,6]

1 LVMT, Université Gustave Eiffel, Ecole des Ponts, Champs-sur-Marne, France, 2 Université Paris Nanterre, UMR 7533 LADYSS, Nanterre, France, 3 Université Paris-Est, Lab'Urba, France, 4 Institut Universitaire de France (IUF), Paris, France, 5 Hôpital Privé des Peupliers, Ramsay Santé, Paris, France, 6 Centre de recherche des Cordeliers, Sorbonne Université, Université de Paris, INSERM, Equipe Etres, France, 7 CEMKA, Paris, France

* benoit.conti@univ-eiffel.fr

**Data Availability Statement:** All relevant data are within the paper and its Supporting Information files.

## Abstract

Socio-economic and geographical inequalities in breast cancer mortality have been widely described in European countries and the United States. To investigate the combined effects of geographic access and socio-economic characteristics on breast cancer outcomes, a systematic review was conducted exploring the relationships between: (i) geographic access to healthcare facilities (oncology services, mammography screening), defined as travel time and/or travel distance; (ii) breast cancer-related outcomes (mammography screening, stage of cancer at diagnosis, type of treatment and rate of mortality); (iii) socio-economic status (SES) at individuals and residential context levels. In total, n = 25 studies (29 relationships tested) were included in our systematic review. The four main results are: The statistical significance of the relationship between geographic access and breast cancer-related outcomes is heterogeneous: 15 were identified as significant and 14 as non-significant. Women with better geographic access to healthcare facilities had a statistically significant fewer mastectomy (n = 4/6) than women with poorer geographic access. The relationship with the stage of the cancer is more balanced (n = 8/17) and the relationship with cancer screening rate is not observed (n = 1/4). The type of measures of geographic access (distance, time or geographical capacity) does not seem to have any influence on the results. For example, studies which compared two different measures (travel distance and travel time) of geographic access obtained similar results. The relationship between SES characteristics and breast cancer-related outcomes is significant for several variables: at individual level, age and health insurance status; at contextual level, poverty rate and deprivation index. Of the 25 papers included in the review, the large majority (n = 24) tested the independent effect of geographic access. Only one study explored the combined effect of geographic access to breast cancer facilities and SES characteristics by developing stratified models.

**Funding:** - CN - Institut National Du Cancer (INCA) - https://www.e-cancer.fr/ - The funders had no role in study design, data collection and analysis, decision to publish, or preparation of the manuscript.

**Competing interests:** The authors have declared that no competing interests exist.

## 1. Introduction

In 2020, breast cancer is the most common cancer among women, with an estimated 685,000 deaths worldwide according to the International Agency for Research on Cancer (IARC). Socio-economic and geographical inequalities in breast cancer mortality have been widely described in European countries and the United States [1–3]. Until the 1970s, although breast cancer incidence was higher among women with a high educational level, their overall survival rate was better than women with a low educational level. The higher incidence among women with a high educational level is currently diminishing with higher rates observed among the most disadvantaged groups [4, 5]. In France, studies show that women with a low SES have lower geographic access to screening mammography than women with a high SES, which is one of the causes of late diagnosis [6]. In the United Kingdom (UK), breast cancer in patients of low socioeconomic status (SES) is more likely to be diagnosed at an advanced stage than in patients with a high SES, leading to lower patient survival [7].

Relationships between breast cancer and social characteristics at contextual level have also been observed [8]. The residents of low SES neighborhoods have, for instance, a significantly lower likelihood of having access to the highest quality of care [9, 10]. In the United States, Yu [11] found that women living in the most socioeconomically disadvantaged areas have a statistically higher risk of dying from cancer. In France, research has shown that the residents of disadvantaged neighborhoods, or rural areas with low medical density, have less access to screening and are diagnosed with more advanced cancer [12].

A systematic review by Khan-Gates et al. [13], has compared the results of 21 studies that examined the relationship between stage of cancer at diagnosis and geographic access to breast cancer screening (mammography). The authors observed that better geographic access to screening facilities was related with greater use of mammography (6 out of 9 relationships) and that better geographic access is related with earlier stage diagnosis (9 out of 22 relationships). However, this review did not examine relationships between geographic access to healthcare facilities according to socioeconomic characteristics to better understand interactions between spatial and social inequalities of breast cancer outcomes. There is therefore a need to explore the combined effects of geographic access and socio-economic characteristics (at individual and contextual levels) on breast cancer outcomes.

This systematic review aims to synthesize the current evidence of relationships between breast cancer outcomes and geographic access according to SES characteristics. In other words, in the context of equal geographic access to healthcare facilities, do women with disadvantaged social and economic characteristics have poorer breast cancer outcomes than more advantaged women? Second, in the case of equal socioeconomic level, do women with poor geographic access to healthcare facilities have worse breast cancer outcomes than women with higher geographic access? To answer these two general questions, the result section will be divided into four research questions: (i) what measures of breast cancer outcomes, geographic access, and SES characteristics? (ii) What are the relationships between geographic access to health-care facilities and breast cancer outcomes? (iii) What are the relationships between SES characteristics and breast cancer outcomes? (iv) What are the combined effects of geographic access and SES characteristics on breast cancer outcomes?

## 2. Method

### 2.1. Literature search strategy

Searches were conducted in PubMed, Web of Science and Scopus using the following Medical Subject Headings (MESH) terms in the title and the abstract. The search was limited to English

language papers that had been published through to April 15, 2019. The following keywords were used for this search:

- ("breast cancer" or "breast neoplasm" or "breast neoplasms" or "breast carcinoma" or "breast tumor" or "breast tumors" or "cancer of the breast")

- **AND** ("accessibility" or "geographic access" or "spatial access" or "residence characteristics" or "residence characteristic" or "neighborhood characteristic" or "neighborhood characteristics")

- **AND** ("SES" or "low-income" or "low income" or "low SES" or "low socioeconomic" or "socioeconomic status" or "low socioeconomic status" or "poor" or "poverty" or "disparity" or "disparities" or "deprived" or "disadvantaged" or "low resources" or "poverty area" or "deprivation" or "social class" or "socioeconomic factors" or "insecurity" or "precariousness").

## 2.2. Inclusion criteria

After excluding duplicate papers, 215 papers were identified by the searches in the three databases. The titles and summaries of these papers were all examined by three reviewers (B.C., A. B. and H.C.). Fig 1 presents the flowchart of the systematic literature search based on PRISMA statement guidelines [14]. The protocol for this literature search was registered in the Prospero database, registration number CRD42020193325 (this can be found at https://www.crd.york. ac.uk/prospero/display_record.php?RecordID=193325).

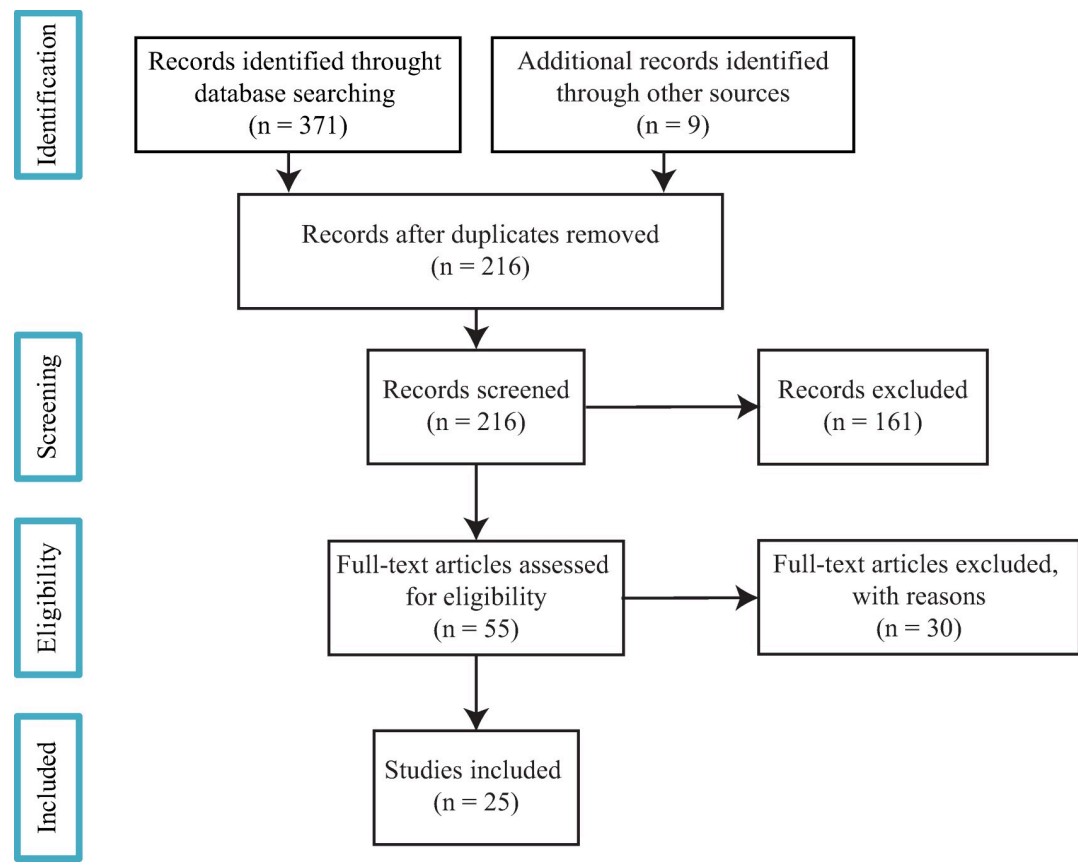

**Fig 1. Flowchart of selection process.**

The selected papers had to meet three major criteria:

i. include at least one of the following measures of breast cancer outcomes: mammography use (Yes/No), stage at diagnosis, type of treatment received such as mastectomy or breast conserving surgery (BCS), and, last, breast cancer survival or mortality;

ii. include socio-economic characteristics at the individual or contextual levels;

iii. include one or more measures of geographic access to healthcare facilities (distance, travel time and geographic capacity).

Based on these criteria, 55 papers were selected.

In the next stage, 30 papers were excluded. We have excluded six papers in which the SES characteristics were demographic such as age or ethnicity [15–19] or used as covariates [20]. Studies were also excluded if they provided only descriptive analyses of level of access to healthcare facilities or neighborhood SES characteristics (10 papers). We excluded one paper that focused on the likelihood of not using the closest facilities [21]. Furthermore, we excluded 13 papers that used a *proxy* measure of geographic access such as density (for example as a measure of healthcare availability), car ownership or urban contexts (urban or rural).

After this elimination process, 25 papers were included in the review. Any papers for which inclusion was open to question were discussed by all the authors until consensus was reached.

## 2.3. Data extraction

For the 25 selected papers, several items of data were extracted by the primary reviewer (B.C.) and presented in an Excel spreadsheet. These were: authors, year of publication, geographical area (country, state or city), breast cancer-related outcomes, geographic access measures and relationships, SES characteristics and relationships.

## 2.4. Quality assessment of included studies

For quality assessment, we adapted the Effective Public Health Practice Project (EPHPP) quality assessment tool [22]. This tool is widely used to evaluate any quantitative study design. We have kept five in the eight key domains for assessment of study quality (study design, selection bias, confounders, data collection and data analysis) according to the study design of studies included. An overall rating for each study was determined based on the component ratings, ranging from 1 (low risk-of bias; high methodological quality) to 3 (high risk-of-bias; low methodological quality). Strong was attributed to those with no weak ratings and at least two strong ratings, moderate was given to those with one weak rating or fewer than two strong ratings and weak was attributed to those with two or more weak ratings. The methodological quality assessment of each of the included studies was independently assessed by three authors (BC, AB and HC). The ratings for each of the five domains, as well as the total rating, were compared between the three authors. Consensus was reached on a final rating for each included article.

## 3. Results

The papers we examined were published between 2002 and 2019, mainly in the latter part of the period between 2009 and 2019 (n = 21). Table 1 sets out the characteristics of the 25 papers. Overall, for 16 articles the methodological quality was rated as strong, for 7 articles as moderate and for 2 as weak (full details on the quality assessment are provided in additional S1 File). Most of the studies were conducted in the United States (n = 18). Three were conducted in Australia, specifically in the state of Queensland [23–25], two in the United Kingdom [26, 27],

**Table 1.** Characteristics of the 25 papers included in the review.

| Study Author, Publication Date | Country, State, City | Sample size | Outcome | Geographic access | | | Transport mode | Relationship Odds ratio [IC] or coefficients | Characteristics at (i) Individual level (r) Residential level | Quality assessments |
| | | | | Measures | From | To | | | | |
|---|---|---|---|---|---|---|---|---|---|---|
| Baade *et al.*, 2016 | Australia, Queensland | n = 11,631 | Treatment (BCS vs mastectomy) | Travel time | Statistical Local Area (SLA) centroid | Closest radiation facility | Car | Less access -> more mastectomy 1h: 1 (ref) 1-2h: 0.58 [0.49–0.69] 2-6h: 0.47 [0.41–0.54] 6h+: 0.44 [0.34–0.56] | • Age (i) • Partner status (i) • Residential disadvantage (r) | 1 |
| Celaya *et al.*, 2010 | USA, New Hampshire | n = 5,966 | Stage at diagnosis | Distance Travel time | Street address (91.5%), or Zip code centroid (8.5%) | Closest mammography facility | Car | NS (not significant) | • Age (i) • Partner status (i) • Health insurance (i) | 1 |
| Dai, 2010 | USA, Michigan, Detroit | n = 12,413 | Stage at diagnosis | Capacity | ZIP code population • weighted centroid | Health care facility | Car | Less access -> more late stage Coefficient of mammography access: -0.191 | Factor analysis on 14 variables (r) • Black population • Black residential segregation • Carless household • Unemployed population (16+) • Female headed household • Population (17+) in poverty • Occupied home ownership • Professional and managerial occupations • Median household income • Median housing value • Median gross rent • Population without a high-school degree • Linguistically isolated household • Household with more than one occupant per room | 1 |

*(Continued)*

Table 1. (Continued)

| Study Author, Publication Date | Country, State, City | Sample size | Outcome | Geographic access | | | | | Characteristics at (i) Individual level (r) Residential level | Quality assessments |
|---|---|---|---|---|---|---|---|---|---|---|
| | | | | Measures | From | To | Transport mode | Relationship Odds ratio [IC] or coefficients | | |
| Dasgupta et al., 2016 | Australia, Queensland | n = 4,104 | Treatment (Breast reconstruction vs "mastectomy only") | Travel time | Statistical Local Area (SLA) centroid | Closest radiation facility | Car | Less access -> more mastectomy 2h: 1 (ref) 2-6h: 0.73 [0.54–0.95] 6h+: 0.26 [0.13–0.59] | • Age (i) • Partner status (i) • Ethnicity (i) • Nationality (i) • Residential disadvantage (r) | 2 |
| Dasgupta et al., 2017 | Australia, Queensland | n = 38,706 | Stage at diagnosis | Travel time | Statistical Local Area (SLA) centroid | Closest radiation facility | Car | Less access -> more late stage 2h: 1 (ref) 2-6h: 0.99 [0.94–1.07] 6h+: 1.18 [1.09–1.28] | • Age (i) • Partner status (i) • Ethnicity (i) • Residential disadvantage (r) | 1 |
| Engelman et al., 2002 | USA, Kansas | n = 117,901 | Mammography screening | Distance | Zip code centroid | • Closest mammography facility • Closest mobile mammography | Car | NS | • Age (i) • Ethnicity (i) • % residents with a high school education (r) | 1 |
| Goovaerts, 2010 | USA, Michigan | n = 2,118 | Stage at diagnosis | Distance | Census tract centroid | Closest clinic | Euclidian Distance | NS | • Census-tract poverty level (r) | 3 |
| Henry et al., 2013 | USA, Arkansas, California, Idaho, Iowa, Kentucky, New Hampshire, New Jersey, New York, North Carolina, Oregon | n = 161,619 | Stage at diagnosis | Travel time Capacity | Population-weighted centroid of census tract | • Closest FDA certified mammography facility • Facilities within a given drive-time catchment area | Car | NS (travel time) NS (capacity) | • Census tract poverty (r) | 1 |
| Henry et al., 2011 | USA, Arkansas, California, Iowa, Idaho, Kentucky, North Carolina, New Hampshire, New Jersey, New York, Oregon | n = 161,619 | Stage at diagnosis | Travel time | Census tract centroid Residential address | • Closest mammography facility • Diagnosing facility | Car | NS (closest facility) Less access to diagnosing -> less late stage 10min: 1 (ref) 10-20min: 0.95 [0.92–0.97] 20-30min: 0.96 [0.92–0.99] 30-40min: 0.98 [0.93–1.03] 40-50min: 0.83 [0.78–0.89] 50-60min: 0.96 [0.87–1.05] 60min+: 0.88 [0.82–0.94] | • Age (i) • Ethnicity (i) • Insurance status (i) • Census tract poverty (r) | 1 |

(Continued)

**Table 1.** (Continued)

| Study Author, Publication Date | Country, State, City | Sample size | Outcome | Geographic access | | | | | Characteristics at (i) Individual level (r) Residential level | Quality assessments |
|---|---|---|---|---|---|---|---|---|---|---|
| | | | | Measures | From | To | Transport mode | Relationship Odds ratio [IC] or coefficients | | |
| Henry et al., 2014 | USA, Utah | n = 5,197 | Mammography screening | Travel time Capacity | Block group population weighted centroid | Closest facility | Car | NS | • Age (i)<br>• Partner status (i)<br>• Ethnicity (i)<br>• Health insurance (i)<br>• Education level (i)<br>• Income (i)<br>• Number of dependent children (i) | 1 |
| Huang et al., 2009 | USA, Kentucky | n = 12,322 | Stage at diagnosis | Distance | Residential address (78%) or Zip code centroid (22%) | Closest mammogram facility | Car | Less access -> more late stage 0-5m: 1 (ref) 5-9m: 1.02 [0.88–1.18] 10–14m: 1.09 [0.91–1.31] 15+m: 1.50 [1.25–1.80] | • Age (i)<br>• Ethnicity (i)<br>• Health insurance (i)<br>• Education level (i)<br>• Education level of census tract (r) | 1 |
| Jones et al., 2008 | UK, Northern England | n = 28,002 | • Survival<br>• Stage at diagnosis | Travel time | Residential address | Closest cancer centre | Car | Less access -> lower survival Travel time to first hospital (min): 0.955 [0.993–0.997] Less access -> more late stage Travel time to GP surgery (min): NS | • Age (i)<br>• Residential disadvantage (r) | 2 |
| Kim et al., 2013 | USA, Illinois, Cook county | n = 21,085 | Stage at diagnosis (normal vs abnormal mammogram) | Distance | Residential address | Actual clinic where women obtained a mammogram | Car | Less access -> more abnormal mammogram Distance in miles: 1.06 | • Age (i)<br>• Partner status (i)<br>• Ethnicity (i)<br>• Education (i)<br>• Income (i)<br>• Employment status (i)<br>• Poverty ratio (r)<br>• African Americans ratio (r) | 1 |

(Continued)

Table 1. (Continued)

| Study Author, Publication Date | Country, State, City | Sample size | Outcome | Geographic access Measures | From | To | Transport mode | Relationship Odds ratio [IC] or coefficients | Characteristics at (i) Individual level (r) Residential level | Quality assessments |
|---|---|---|---|---|---|---|---|---|---|---|
| Lian et al., 2012 | USA, Missouri, St. Louis City and St. Louis County | n = 4,205 | Stage at diagnosis | Travel time, Capacity | Block group population weighted centroid | • Closest mammography facilities • Five closest facilities • Total mammography facilities that can be reached within 30 minutes | Car | NS (closest facility) NS (total mammography facilities) Spatial accessibility index: 1.19 [1.03–1.37] | SES deprivation index based on 9 variables (r) • % civilian labor force unemployed • % vacant household • % household with >= 1 person per room • % female headed household with dependent children • % household on public assistance income • % household with no vehicle • % household with no phone • % population below federal poverty line • % non-Hispanic (NH) African Americans | 1 |
| Lin et al., 2018 | USA, South Dakota | n = 4,031 | Treatment (mastectomy vs BCS) | Travel time | Residential address | Closest radiotherapy facility | Car | Less access -> more mastectomy 0-30min: 1 (ref) 30-60min: 1.06 [0.80–1.41] 60-90min: 1.30 [0.95–1.77] 90-120min: 1.51 [1.08–2.11] 120min+: 1.70 [1.119–2.42] | • Age (i) • Ethnicity (i) • Poverty rate (r) | 1 |
| Lin and Wimberly, 2017 | USA, South Dakota | n = 6,418 | Stage at diagnosis | Capacity | Census-tract centroid | • Closest mammography facilities • Closest primary care physicians | Car | NS | • Age (i) • Ethnicity (i) • Residential deprivation (r) | 1 |

(Continued)

**Table 1.** (Continued)

| Study Author, Publication Date | Country, State, City | Sample size | Outcome | Geographic access | | | | Relationship Odds ratio [IC] or coefficients | Characteristics at (i) Individual level (r) Residential level | Quality assessments |
|---|---|---|---|---|---|---|---|---|---|---|
| | | | | Measures | From | To | Transport mode | | | |
| McLafferty et al., 2011 | USA, Illinois | n = 37,392 and n = 44,070 | Stage at diagnosis | Capacity | ZIP code population-weighted centroid | Primary health care physicians | Car | Less access -> more late stage Capacity: -37.092 | • Age (i)<br>• Ethnicity (i)<br>Factor analysis on 11 variables (r)<br>• population with high healthcare needs<br>• population in poverty<br>• female-headed households<br>• home ownership<br>• median income<br>• households' density<br>• with an average of more than one person per room and<br>• housing units that lack basic amenities<br>• nonwhite population<br>• population without a high-school diploma<br>• households linguistically isolated<br>• households without vehicles | 2 |
| Onitilo et al., 2013 | USA, Wisconsin, Marshfield | n = 1,368 | Mammography screening | Travel time | Residential address | Closest mammography center | Car | Less access -> less mammography Time (minutes): 0.990 [0.986–0.993] | • Age (i)<br>• Health insurance (i) | 3 |
| Rocha-Brischiliari et al., 2018 | Brazil, Parana state | n = 2,215 | Survival/mortality | Capacity | Municipality of residence centroid | • Closest radiotherapy facility oncological service<br>• Reference services located within the catchment of each Area | Car | More access -> more mortality Capacity: 12.9527 | • Illiteracy level (r)<br>• Per capita income (r) | 2 |
| Sauerzapf et al., 2008 | UK, Northern England | n = 6,014 | Treatment (BCS vs mastectomy) | Travel time | Residential postcodes centroid | Closest radiotherapy facility | Car | NS | • Age (i)<br>• Residential deprivation (r) | 2 |

(*Continued*)

Table 1. (Continued)

| Study Author, Publication Date | Country, State, City | Sample size | Outcome | Geographic access | | | | Relationship Odds ratio [IC] or coefficients | Characteristics at (i) Individual level (r) Residential level | Quality assessments |
|---|---|---|---|---|---|---|---|---|---|---|
| | | | | Measures | From | To | Transport mode | | | |
| Schroen and Lohr, 2009 | USA, Virginia | n = 8,170 | Stage at diagnosis | Distance | Residential address | Closest mammography facility | Car | NS | • Age (i) • Ethnicity (i) • Per capita income (r) | 2 |
| St-Jacques et al., 2013 | Canada, Quebec | n = 833,856 | Mammography screening | Distance | Residential address | Closest designated screening centre | Car | Less access -> less mammography 0–2.5km: 1 (ref) 2.5–5km: 1 [1.00–1.01] 5–12.5km: 1 [0.99–1.00] 12.5–25km: 0.96 [0.96–0.97] 25–50km: 0.96 [0.95–0.97] 50–75km: 0.88 [0.86–0.89] 75km+: 0.81 [0.79–0.83] | • Age (i) Material deprivation index (r) • % with no high school diploma • employment/ population ratio • average income Social deprivation index (r) • % of respondents living alone • % of individuals separated divorced or widowed • % of single-parent families | 1 |
| Tarlov et al., 2009 | USA, Illinois, Chicago | n = 4,533 | Stage at diagnosis | Distance | Residential address (94%) or Zip code centroid (6%) | Closest five facilities | Car | NS | • Age (i) • Ethnicity (i) • Poverty rate (r) • High school graduation rate (r) | 1 |
| Voti et al., 2006 | USA, Florida | n = 18,903 | Treatment (BCSR vs Mastectomy) | Distance | Residential address (98%) | Closest radiotherapy facility | Euclidean distance | Less access -> more mastectomy per 5 mile increase: 0.97 [0.95–0.99] per 10 mile increase: 0.94 [0.90–0.98] per 15 mile increase: 0.91 [0.86–0.96] per 20 mile increase: 0.88 [0.82–0.95] | • Age (i) • Partner status (i) • Ethnicity (i) • Health insurance (i) | 1 |

(Continued)

**Table 1.** (Continued)

| Study Author, Publication Date | Country, State, City | Sample size | Outcome | Geographic access | | | | Relationship Odds ratio [IC] or coefficients | Characteristics at (i) Individual level (r) Residential level | Quality assessments |
|---|---|---|---|---|---|---|---|---|---|---|
| | | | | Measures | From | To | Transport mode | | | |
| Yang & Wapnir, 2018 | USA, California, Stanford University | n = 1,938 | Treatment (Breast conservation, Unilateral mastectomy, Bilateral mastectomy or Postmastectomy) | Distance | Zip code centroid | Address of the hospital | Car | NS | • Age (i)<br>• Partner status (i)<br>• Health insurance (i) | 2 |

BCS: Breast Conserving Surgery; BCSR: Breast-conserving surgery with radiation; (%): percentage of women geolocalized at residential address or at area level (zip code centroid)

Quality assessment rating: 1 (strong), 2 (moderate), 3 (weak)

one in Brazil [28] and one in Canada [29]. The study area ranged from several federated states to an individual hospital. The most frequent study scale was the federated state (or region). This was the case for 20 papers, and 4 of them included more than one region. Three studies were conducted at the city level [30–32] and two at the hospital level [33, 34].

## 3.1. What measures of breast cancer outcomes, geographic access, and SES characteristics?

The most explored outcome was the stage of cancer at diagnosis (n = 15), followed by the probability of the women receiving different types of treatment such as mastectomy, breast conserving surgery and/or radiotherapy (n = 6). Screening mammography was assessed in three papers and breast cancer mortality in two (Table 1). One paper explored two breast cancer-related outcomes: cancer stage at diagnosis and survival [26].

Using Geographical Information Systems (GIS), geographic access between the residential address of the women and the closest healthcare facility was evaluated by travel time (12 papers) or/and by travel distance (10 papers) or/and by the two-step floating catchment area method (2SFCA) (6 papers). This method is based on the results of spatial capacity modelling including population demand and healthcare provision. Three studies combined two measures such as travel time and capacity [35, 36] or travel distance and travel time [37]. Geographic measures were assessed by Euclidean distance (n = 2) or/and by car (n = 23). In the large majority of the studies, the travel time or distance was estimated between the closest healthcare facility and the women's residential addresses (n = 11) or the centroid of their residential neighborhood (n = 18). Only two papers calculated the distance between the healthcare

**Table 2. Cut-off of travel distance and travel time of included articles.**

| | Cut-off | |
|---|---|---|
| | **Travel distance** | **Travel time** |
| St-Jacques *et al.*, 2013 | <2.5; 2.5–5; 5–12.5; 12.5–25; 25–50; 50–75; >75 (km) | - |
| Huang *et al.*, 2009 | <5; 5–10; 10–15; >15 (mi.) | - |
| Engelman *et al.*, 2002 | <5; 5–10; 10–20; >20 (mi.) | - |
| Yang & Wapnir, 2018 | <10; 10–30; 30–60; >60 (mi.) | - |
| Tarlov *et al.*, 2009 | Continuous value | - |
| Kim *et al.*, 2013 | Continuous value | - |
| Henry *et al.*, 2013 | - | <5; 5–10; 10–20; 20–30 (min) |
| Henry *et al.*, 2011 | - | <10; 10–20; 20–30; 30–40; 40–50; 50–60 (min) |
| Henry *et al.*, 2014 | - | <20; >20 (min) |
| Sauerzapf *et al.*, 2008 | - | <30; 30–60; >60 (min) |
| Lin *et al.*, 2018 | - | <30; 30–60; 60–90; 90–120; >120 (min) |
| Baade *et al.*, 2016 | - | <1; 1–2; 2–6; >6 (hr) |
| Dasgupta *et al.*, 2016 | - | <2; 2–6; >6 (hr) |
| Dasgupta *et al.*, 2017 | - | <2; 2–6; >6 (hr) |
| Jones *et al.*, 2008 | - | Continuous value |
| Schroen and Lohr, 2009 | - | Continuous value |
| Onitilo *et al.*, 2013 | - | Continuous value |
| Celaya *et al.*, 2010 | <5; 5–10; 10–15; >15 (mi.) | <5; 5–10;>15 (min) |

mi.: miles; km: kilometers; min: minutes; hr: hours.

facilities used by the women and their residential addresses [34, 38]. In addition, the cut-off used to categorize travel distances and travel times varied (Table 2). Celaya *et al.* [37] proposed three classes of travel time in which the least accessible class was ">15min"; whereas in Sauerzapf et al. [27] proposed a three-classes in which the most accessible class was "<30min" and the least accessible class was ">60min".

Socioeconomic characteristics were assessed at individual and residential levels: fifteen studies combine data from both levels, five papers present only individual level data and five papers use data at residential level only. At individual level, the most common characteristics used were age (n = 20), ethnicity (n = 13), partner status (n = 8), health insurance (n = 7) and education level (n = 2). At residential level, the main data used were residential disadvantage or deprivation (n = 10): six papers adopted existing deprivation indicators (e.g., Index of relative socioeconomic Advantage and Disadvantage [IRSAD], index of multiple deprivation [IMD]), and four papers created their own deprivation index [29, 31, 32, 39]. Other variables used were poverty (n = 5), educational level (n = 4) and per capita income (n = 2).

### 3.2. What are the relationships between geographic access to health-care facilities and breast cancer outcomes?

The twenty-nine relationships between geographic access and at least one breast cancer outcome (mammography use, stage at diagnosis, treatment and mortality) explored in the 25 papers in our review are presented in Table 3. In overall, the statistical significance of the relationships was heterogeneous with 15 significant relationships and 14 as non-significant.

Women with high level of geographic access to healthcare facilities had a statistically significant higher cancer screening rate in one study (to 4), an earlier stage of cancer at diagnosis

**Table 3. Relations between breast cancer outcomes and geographic access to health-care facilities.**

| | Geographic access measures | | |
|---|---|---|---|
| | **Travel time** | **Travel distance** | **Capacity** |
| Mammography use | Henry *et al.*, 2014 (NS) | Engelman *et al.*, 2002 (NS) <br> St-Jacques *et al.*, 2013 (+) | Henry *et al.*, 2014 (NS) |
| Stage at diagnosis | Celaya *et al.*, 2010 (NS) <br> Henry *et al.*, 2013 (NS) <br> Henry *et al.*, 2011 (NS) <br> Dasgupta *et al.*, 2017 (+) <br> Onitilo *et al.*, 2013 (+) <br> Jones *et al.*, 2008 (+) | Celaya *et al.*, 2010 (NS) <br> Tarlov *et al.*, 2009 (NS) <br> Goovaerts, 2010 (NS) <br> Schroen and Lohr, 2009 (NS) <br> Huang *et al.*, 2009 (+) <br> Kim *et al.*, 2013 (+) | Lin and Wimberly, 2017 (NS) <br> Henry *et al.*, 2013 (NS) <br> Dai, 2010 (+) <br> Lian *et al.*, 2012 (+) <br> McLafferty *et al.*, 2011 (+) |
| Treatment | Sauerzapf *et al.*, 2008 (NS) <br> Baade *et al.*, 2016 (+) <br> Dasgupta *et al.*, 2016 (+) <br> Lin *et al.*, 2018 (+) | Yang & Wapnir, 2018 (NS) <br> Voti *et al.*, 2006 (+) | |
| Survival/mortality | Jones *et al.*, 2008 (-) | | Rocha-Brischiliari *et al.*, 2018 (-) |

NS: not significant

+: better geographic access related with better breast cancer-related outcomes (higher screening rate, early stage, fewer mastectomies, lower mortality rate)

-: better geographic access related with poorer breast cancer-related outcomes (lower screening rate, late stage, more mastectomies, higher mortality rate)

The travel distance is the distance between the women's residential addresses or the centroid of their neighborhood and their healthcare facility

Travel time is the time taken to travel between the women's residential addresses or the centroid of their neighborhood and their healthcare facility

Capacity: spatial modelling based on population demand and healthcare provision

(n = 8/17), and fewer mastectomies (n = 4/6) than women with lower level of geographic access. Two studies with survival rates as outcomes observed that women with higher access to healthcare facilities have poorer survival rates than women with lower access to healthcare [26, 28]. The authors of the UK study put forward that this result may be "an artefact of imperfect control of the effects of deprivation, since inner city populations tend to be more deprived and closer to hospitals than suburban or rural populations" [26]. In the state of Parana in Brazil, the authors suggested that the municipalities close to the services of specialized treatment in oncology were also areas with high population concentration which makes access to treatment difficult [28].

As reported in Table 3, relationships varied considerably depending on the breast cancer outcomes: geographic access seems to more frequently influence the type of treatment (4/6) than whether women undergo screening (1/4). It therefore seems necessary to explore whether the type of measure of geographic access (travel time, travel distance or capacity in Table 3) influences the results of the relationships and is responsible for the differences observed between the included studies.

As presented in Table 3, several measures of geographic access were used, which raises the question of their impact on the relationships (significance and direction) with breast cancer outcomes. When geographic access measures were based on travel times, the results varied: 7 studies found negative relationships and 4 found no relationship [27, 35–37, 40]. When geographic access was measured by travel distance, a non-significant relationship was observed in 6 studies [30, 34, 37, 41–43] and a significant one in 4 studies [29, 38, 44, 45]. Finally, when modelling was applied to combine demand and travel time to healthcare (the 2SFCA method), 4 papers observed a significant relationship with breast cancer outcomes [28, 31, 32, 39] and 3 observed a non-significant relationship [35, 36, 46].

Regardless of the type of measures used to calculate geographic access to health facilities (distance, time or geographical capacity), the heterogeneity of the results is very similar and does not allow to define which proxy is the most appropriate to assess geographic access. This finding is confirmed by the fact that studies that compared two different measures of geographic access obtained similar results [35–37]. For example, the paper by Celaya *et al.* [37] compared travel distance and travel time by car between patients' addresses and a mammography service in the state of New Hampshire, USA. For these two measures, the authors showed an absence of a relationship between geographic access and stage at diagnosis.

Another interesting finding is that the heterogeneity of the results regarding a relationship between geographic access and breast cancer outcomes cannot be explained by the geographical context and, in particular, the country of study: for example, among 16 studies conducted in the USA, 8 found significant relationships and the other 8 found non-significant relationships. In addition, neither the nature of the area studied (e.g. metropolitan, urban area, rural area) nor the geographic level (e.g. federal state, city, hospital) appear to affect the ability of the studies to explain differences in breast cancer outcomes.

## 3.3. What are the relationships between SES characteristics and breast cancer outcomes?

SES characteristics were explored at the individual level in 5 papers, at the residential level in 5 papers, and at both the individual and residential levels in 15 papers. At the individual level, SES characteristics were mostly assessed by partner status and health insurance status. Based on Table 1, age and tumor features could be assessed as major confounding variables in the relationships between SES, geographic access and breast cancer outcomes. At the residential level, SES was defined by the poverty rate or the income level (8 papers) or by composite scores

such as the deprivation index (10 papers). The composite scores were based on different variables, statistical methods and geographical scales. For instance, Lian *et al.* [32] used 9 variables in a multivariate approach in order to define a deprivation index, whereas St-Jacques *et al.* [29] defined two indices (material deprivation and social deprivation), using a factor analysis based on 3 variables in each case.

At the individual level, as shown on the forest plot (Fig 2), the relationships between breast cancer outcomes and age (a), ethnic status (b) and socioeconomic status were mixed (c and d). Considering the studies as a whole, the relationships with age did not follow the same pattern. Greater age was related with: (i) more mammography use (except for 1 study [36]); (ii) lower odds of late stage at diagnosis (except for 3 studies [26, 30, 38]); (iii) receiving Breast-Conserving Surgery (BCS) (except for 1 study [34]). The relationships with ethnic origins were also not systematic and were based on many different definitions of ethnic origins which made it difficult to compare the studies. The majority of the relationships between marital status and breast cancer outcomes were not significant [36, 38, 44]. In two studies, married women or with partner had lower risks of late-stage diagnosis of breast cancer than other women [25, 37]. In two papers, married women or with partner had higher odds of receiving BCS than single women [23, 44]. According to the authors of these studies, these findings reflect the underlying issue of social support networks and social incentives which may affect women's motivation or ability to screen and/or to receive BCS. In the six papers, the majority of the relationships between marital status and breast cancer outcomes were not significant (6 relationships). Women with no health insurance had less mammography screening and more advanced cancer stage at diagnosis than women with health insurance. The relationship between health insurance and receiving Breast-Conserving Surgery (BCS) seems to be less significant.

At the residential level, as reported in six papers (Fig 2E), women residing in residential environments characterized by high levels of poverty were more prone to late-stage diagnosis than the others (except for two studies [38, 41]). The relationship with the type of treatment received was not significant except in one relationships [47], and in this case only for women residing in an area with a very high level of poverty ($> = 15\%$). In contrast, the relationship between deprivation index (Fig 2F) at residential level and type of treatment was identified as significant in 3 studies: women who resided in the most disadvantaged areas seemed to undergo less BCS [23] and more mastectomies [24, 27] than the others. The deprivation index exhibited no consistent relationship with stage at diagnosis. Only two studies investigated the relationships with the use of mammography or survival rates: living in an area with a high level of material or social deprivation was related with lower mammography use [29], later stage presentation and higher mortality risk [26].

### 3.4. What are the combined effects of geographic access and SES characteristics on breast cancer outcomes?

Of the 25 papers included in the review, the large majority (n = 24) tested the independent effect of geographic access. In these studies, SES characteristics were used as predictors and/or covariates. Only one study explored the combined effect of geographic access to breast cancer facilities and SES characteristics on the probability of different breast cancer outcomes. Lian *et al.* [32] developed stratified models of the effects of geographic access to mammography services and neighborhood socio-economic deprivation on late-stage breast cancer diagnosis. The models show that lower geographic access to mammography services was related with greater odds of late-stage breast cancer diagnosis in less deprived neighborhoods, but not in more deprived neighborhoods.

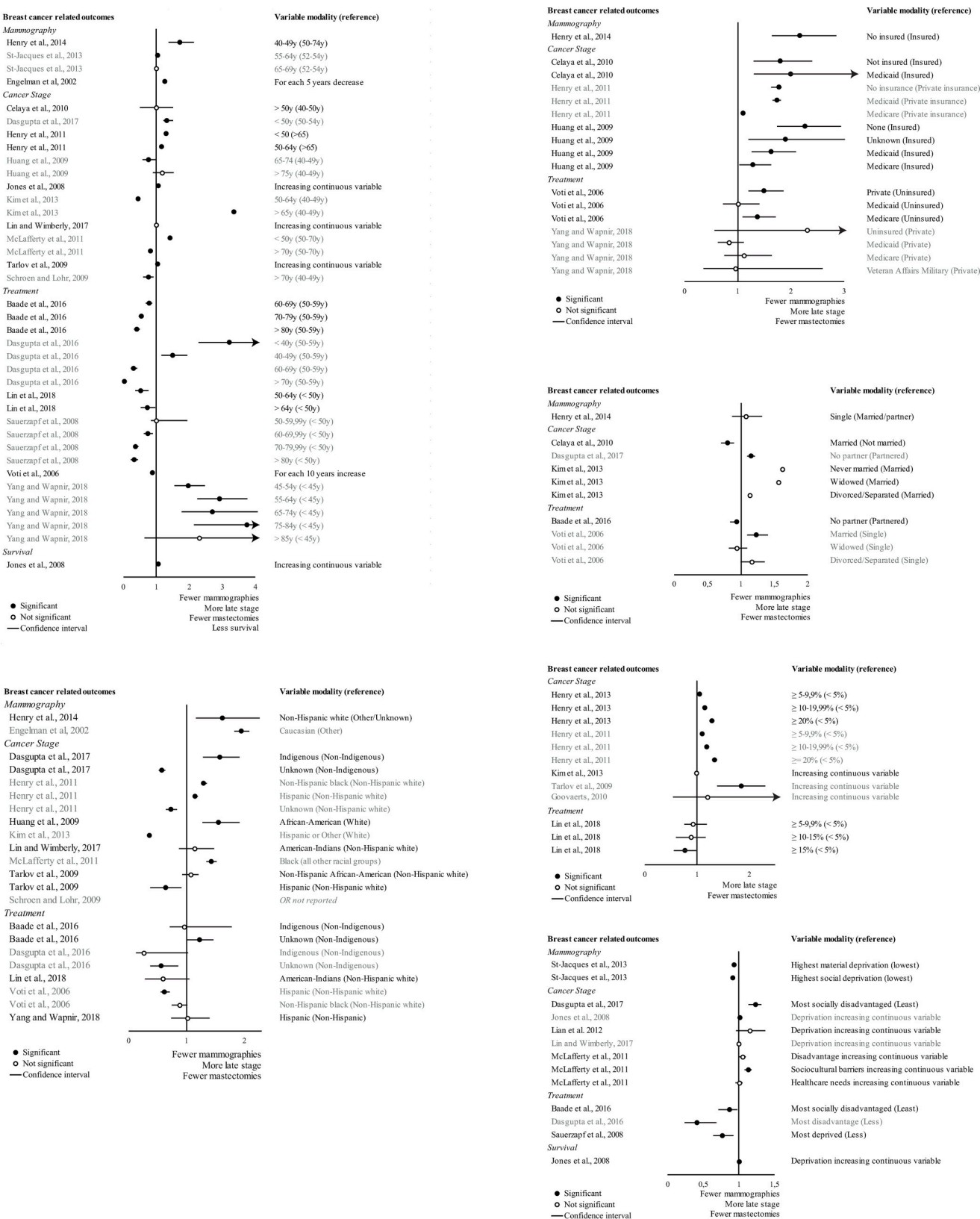

**Fig 2. Forest plot showing the relationship between SES characteristics (at the individual and contextual levels) and breast cancer outcomes.** a) Age, b) Ethnicity, c) Insurance status, d) Marital status, e) Poverty rate at residential level, f) Deprivation index at residential level.

## 4. Discussion

In this review, we have investigated 25 papers that reported 29 relationships between geographic access and breast cancer outcomes according to SES characteristics at the individual and/or residential levels. Three types of measures were used to assess geographic access to the closest facility (travel time or distance and geographical capacity) and four main breast cancer outcomes (mammography use, stage at diagnosis, type of surgical treatment and mortality) were considered. Even if the relationships between geographic access and breast cancer-related outcomes were inconsistent, we observed interesting findings based on a large majority of strong quality studies. First, the type of treatment (Breast-Conserving Surgery [BCS] vs Mastectomy) undergone by women differs significantly according to geographic access. A first hypothesis is that: BCS requires several visits to radiation therapy facilities, whereas mastectomy does not require regular round trips to facilities. A second hypothesis is the lack of information towards women in less accessible areas who therefore do not have the opportunity to make an informed decision regarding the choice of treatment. In addition, the level of specialization, the volume of surgery and/or the type of hospital (private/public) can influence surgical treatment. For instance, Dasgupta et al. (2016) showed that women who have significantly more likely to undergo breast reconstruction (following mastectomies) attended high-volume or private hospitals (and were younger, diagnosed more recently, had smaller tumors, lived in less disadvantaged or more accessible areas).

On the contrary, geographic access does not seem to be a significant determinant of participation in breast cancer screening unlike socioeconomic level. The way geographic access is measured might explain the absence of relationship: in most cases, the distance is assessed between the place of residence of the women and the closest health facility which may appear restrictive to understand the space of sociability of women in a broader way. To overcome this limit, it would be useful to include other places than home such as the workplace and any frequently visited places as well as to analyze women health seeking behaviors including characteristics of the hospital used. In the text below, we propose to explore different directions for further research on the geographic and socioeconomic determinants of breast cancer outcomes.

### Taking account of the fact that people do not necessarily use the closest facility, and that they do not necessarily start from home

The reviewed studies assumed that women had access to and used the closest facility to their home and that the starting point was always their home address or the polygon centroid of the residential area (when the home address was unavailable). The calculated travel time or distance was therefore the minimum possible time or distance. This may differ from the actual travel time or distance based when women do not use the health facilities that are nearest to their home. Alford-Teaster *et al.* [48] have shown that only 35% of women participating in the US-based Breast Cancer Surveillance Consortium in the years 2005–2012 (n = 646,553 women) used their closest mammography facility. In this sample, nearly three-quarters of women not using their closest facility used a facility within 5 minutes of it. A previous study has compared self-reported and calculated measures (by women and GIS respectively) of travel time to the maternity unit for childbirth [49]. The reported travel times were similar to the calculated travel times in peri-urban and rural areas, but agreement between the two was poor in urban areas. To overcome this limitation of theoretical accessibility, future studies will ensure that women's actual care pathways are taken into account including information about their reported travel time and their reasons for choosing (or not choosing) certain types of healthcare facilities. As stated by Khan-Gates et al. [13], we need to further explore the "*the actual*

*geographic patterns of seeking care rather than access to the nearest facilities*". To this purpose, information on where women come from when they go to hospital and why they choose (or not choose) their hospital should be included both in questionnaire and interview. Another methodological limitation observed in the reviewed studies concerns the absence of public transport for the calculation of geographic access based only on the travel distance or time by car.

## Taking account of the use of other modes of transportation

As reported by Celaya *et al.* [37], the car is the main mode of transport in the context of assessments of geographic access in breast cancer studies. Measures based on public transportation are rarely used (n = 3/25), and only in terms of supply density [26, 27, 30]: in these three papers, it is the availability of public transport (buses or trains) in the residential area that is measured (line density or the presence of a stop nearby). None of this research measures geographic access, either in time or distance. The lack of analyses based on travel by public transport in breast cancer issues tends to mask the use of other modes of transportation. We may assume that women who are on low incomes and/or who live in deprived areas are more limited in their choice of travel mode, in particular ownership of a private car. For instance, in London, customers who live in the most deprived areas are less likely to use a private car to travel to convenience stores [50]. In contrast, in inner city areas, where the public transport network is dense and effective, public transport can even be faster than the private car. Thus, the availability of effective and convenient public transport (metro, tram, suburban rail, and bus) may be deemed to be a leading driver of women's mobility. Thus, advances in GIS methods and the availability of transport data sets will allow studies to make a more accurate assessment of the geographic access to healthcare facilities by public transport [51].

In addition to the efforts that must be made to improve geographic access measures (not only the closest facility and include public transportation), the quality of the findings also depends on the intersection of geographic access with socio-economic variables.

## Taking account of interactions between geographic access and deprivation at the individual level

Unfortunately, the combined effects of geographic access to breast cancer facilities and SES characteristics on the probability of different breast cancer outcomes have been less frequently explored. For instance, using stratification analyses, Lian *et al.* [32] showed that the significance of the relationship between geographic access to mammography services and stage at diagnosis varied according to the level of deprivation: women who are more deprived and who live in more accessible areas have less access to mammography screening than non-deprived women who have poorer geographic access. In this way, stratification analysis has been used to divide the study population into several strata according to characteristics that may influence health outcomes. This would help answer the questions we posed, but which we were unable to answer, at the beginning of this systematic review: in a context of equal geographic access to health facilities, do socially and economically disadvantaged women have worse breast cancer outcomes than more advantaged women? With equivalent socio-economic status, do women with poor geographic access to healthcare facilities have worse breast cancer outcomes than women with good geographic access? In addition, the measurement of geographic access should also be considered in relation to the local context in which women live. For example, women who live in suburban or rural areas are more willing to travel longer distances than women living in urban centers [29]. To this purpose, information on urban density level of

residential area of women should be collected either from self-reported questionnaires or from spatial databases (e.g., urban, suburban, rural areas).

## Taking account of the variability of urban forms and local contexts

The great variability in the results may also be partly explained by differences between study areas in terms of social organization and urban forms, which result from a complex system of interactions between social, political, economic and cultural dimensions [52]. Although increasing evidence suggests that urban form affects public health [53], the urban and social morphologies of cities are rarely used to explain inequalities in healthcare access. For instance, certain suburbs of cities may exhibit a lower density of health services and transportation provision as well as a higher level of deprivation, while the opposite may apply in others [28]. Inner city areas may either be characterized by distressed housing, abandoned buildings and vacant lots or, in contrast, the highest housing prices in the city: the geographic distribution of the population and services are highly variable. As geographic and social access to healthcare is highly embedded in local contexts, it will always be difficult to draw general conclusions from the evidence. Healthcare access issues need further study in the case of urban environments in which differing public health and planning policy responses are required to meet the varied challenges [54].

## Taking account of changes in the geographic access score and the deprivation level over time: The need for longitudinal analysis

Twenty of our selected papers were cross-sectional and provided a snapshot at a given time (in general over a four-year period) of the relationship between geographic access, SES and breast cancer outcomes. Only five papers considered a longer period of over 10 years. One of the papers [42], is original in that it covers a very long recruitment period of 17 years—women diagnosed during the period 1985–2002. Three of the other papers took the year of diagnosis as an explanatory variable for differences in cancer outcomes [23–25]. These three papers arrived at the same conclusion: the difference between the likelihood of having better treatment [23, 24] or less advanced cancer at diagnosis [25] between women with good access to healthcare facilities and those with poor access, has decreased over time. It would appear that, over time, the level of geographic access (measured in these three papers by the travel time) has become increasingly less significant: at the beginning of the study period, breast cancer outcomes were very different for women with poor geographic access and the others, while at the end of the period the difference between the two groups was smaller. A contrasting view is presented in the paper by McLafferty *et al.* [39]. This is the only study that compares two cohorts of women ten years apart (1988–1992 vs. 1998–2002). The results also show that a change has occurred over time: the impact of geographic access was statistically significant in the recent period but less so in the early 1990s. How can we explain this finding? Have inequalities in geographic access increased? Have screening techniques improved? There are many hypotheses, and a longitudinal analysis that provides a comparison at the individual level of geographic access and deprivation over time would provide a better understanding of the changes that are occurring.

Our systematic literature review has a number of limitations. First, using a quality assessment tool introduced some challenges. There is no consensus as to whether one should judge the representativeness of these characteristics of the study and the quality of the reviewed studies is based on what the authors reported in the paper. The quality assessment may not reflect a low quality of the study but might merely have been a lack of reported detail in the paper. Second, the heterogeneity of sample size, characteristics of the sample and measurement tools

(both access and SES measures) limited the inter-study comparisons. Third, as in any systematic review, it is possible that some eligible studies may have been missed in our search strategy.

## 5. Conclusion

Our study demonstrates the diversity of the relationships between geographic access to healthcare facilities, SES characteristics and breast cancer outcomes. However, these 25 papers do not allow us to conduct a cross-sectional analysis of the combined effects of geographic access and SES variables and therefore do not allow us to say if a disadvantaged woman with good geographic access to facilities has better outcomes than an advantaged woman living a long way from healthcare facilities.

There are several ways in which the design and implementation of cross-analysis research that deals with the level of geographic access for different levels of deprivation in women with breast cancer can be improved. These are: (i) taking account of individual SES characteristics, in particular an individual level deprivation index; (ii) providing a longitudinal analysis of geographic access; (iii) conducting a qualitative analysis of lifestyles, care pathways and mobility capacities would be very valuable. This requires a specific research protocol based on regular questionnaires and/or interviews at different times, from diagnosis to one or more years later, including dimensions of precariousness and ability to travel as well as spatial access to healthcare (e.g. address of their general practitioners (GP), possession of a driving licence, availability of public transport).

The mechanisms underlying relationships between changes in the urban environment (e.g. location of healthcare, transport networks) as well as in individual characteristics (e.g. car ownership, marital status) and breast cancer outcomes are insufficiently studied. Increased understanding of such mechanisms is much needed to clarify the significance and role of specific modifiable geographic and social determinants along putative causal pathways. Increased knowledge in this field would also inform the design and targeting of future interventions which are crucial issues for public health and urban planning policies and for stakeholders. A major issue of future strategies should be to identify deprived patients at an early stage to implement corrective measures and care management adapted to each level of deprivation. These measures could be geographic (such as opening up in low medical density areas) and/or social (systematic referral of patients to social services of the hospital, work on perceptions of the disease and treatment) and/or medical (promoting participation in clinical trials, provide treatment side effects, facilitate access to supportive care).

## Supporting information

**S1 Checklist. PRISMA 2009 checklist.**
(PDF)

**S1 File. Quality assessment results using an adaptive Effective Public Health Practice Project (EPHPP) tool.**
(PDF)

## Author Contributions

**Conceptualization:** Benoit Conti, Audrey Bochaton, Hélène Charreire.

**Funding acquisition:** Charlotte Ngô.

**Investigation:** Benoit Conti, Audrey Bochaton, Hélène Charreire.

**Methodology:** Benoit Conti, Audrey Bochaton, Hélène Charreire.

**Project administration:** Benoit Conti, Audrey Bochaton, Hélène Kitzis-Bonsang, Charlotte Ngô.

**Resources:** Caroline Desprès, Sandrine Baffert.

**Supervision:** Hélène Kitzis-Bonsang, Caroline Desprès, Sandrine Baffert, Charlotte Ngô.

**Validation:** Benoit Conti, Hélène Charreire, Hélène Kitzis-Bonsang, Caroline Desprès, Sandrine Baffert, Charlotte Ngô.

**Visualization:** Benoit Conti, Audrey Bochaton, Hélène Charreire.

**Writing – original draft:** Benoit Conti, Audrey Bochaton, Hélène Charreire.

**Writing – review & editing:** Benoit Conti, Audrey Bochaton, Hélène Charreire.

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
