## [Decision Letter · Decision Letter 0]

9 Aug 2021

PONE-D-21-10021

Influence of geographic access and socioeconomic characteristics on breast cancer outcomes: a systematic review

PLOS ONE

Dear Dr. Conti,

Thank you for submitting your manuscript to PLOS ONE. After careful consideration, we feel that it has merit but does not fully meet PLOS ONE’s publication criteria as it currently stands. Therefore, we invite you to submit a revised version of the manuscript that addresses the points raised during the review process.

We look forward to receiving your revised manuscript.

Kind regards,

Tzai-Hung Wen, Ph.D.

Academic Editor

PLOS ONE

Journal Requirements:

Reviewers' comments:

Reviewer's Responses to Questions

**Comments to the Author**

1. Is the manuscript technically sound, and do the data support the conclusions?

Reviewer #1: Yes

Reviewer #2: Yes

2. Has the statistical analysis been performed appropriately and rigorously? 

Reviewer #1: Yes

Reviewer #2: N/A

3. Have the authors made all data underlying the findings in their manuscript fully available?

Reviewer #1: Yes

Reviewer #2: Yes

4. Is the manuscript presented in an intelligible fashion and written in standard English?

Reviewer #1: Yes

Reviewer #2: Yes

5. Review Comments to the Author

Reviewer #1: This study conducted a systematic review on the relationships between geographic access and socioeconomic status on breast cancer outcome. These results show that the better geographic access to healthcare facilities had a significant fewer mastectomy. But for the residential level SES, the deprivation index exhibited inconsistent relationship with stage at diagnosis. It is recommended to consider the suggestions below before probably publishing.

1. In Section 3.1, the authors could add a part to general discuss the information about socioeconomic characteristics.

2. In Table 1, there is a typo in the column name “Relationship Odds ration [CI] or coefficients,” and for the row Jones et al., 2008, the odds ratio did not fall into the range of 95% CI, for Onitilo et al., 2013, the upper value of 95% CI is lacking of a decimal point, these should be typos.

3. According to Table 1, could the authors conclude the major confounding variables that must be considered for the future studies?

4. For Table 2, the references of travel distance could be sorted by initial letter to let the table more readable.

5. In Line 192-193, can the authors explain the reason why higher access to healthcare facilities have poorer survival rates?

6. In Line 245-247, how did the marriage status affect the odds of diagnosis stage and the odds of receiving BCS?

7. In Line 281-283, could you explain why the less deprived neighborhoods would have greater odds of late-stage breast cancer diagnosis than more deprived neighborhoods?

8. Authors could consider hospital classification as a surrogate to discuss the relationships of geographical accessibility to healthcare facilities.

9. For Conclusion section, can the authors give a recommendation on what kind of specific intervention or modification strategy for the stakeholders could take?

10. Could the authors give a brief summary about which proxy is more suitable in accessing geographic accessibility and what is the limitation of this kind of theme and how can they improve in the future studies?

Reviewer #2: 1. The authors use the meta-analysis method on 25 studies which focus on the Socio-economic and geographical inequalities in breast cancer mortality. Therefore, I like to review this study according to the purpose, contribution, and the limitation.

2. The main purpose of this study is to synthesize the current evidence of relationships between breast cancer outcomes and geographic access according to SES characteristics by using the meta-analysis method. I suggest the authors to list several specified research questions. Those specified questions can make your readers much more understanding the purpose of this study. Actually, this paper has already provided some basic findings on the general results (3.1), Relationship between geographic access and breast cancer-related outcomes (3.2), Relationship between SES characteristics and breast cancer-related outcomes (3.3), and Relationships according to geographic access measures, SES characteristics and breast cancer outcomes (3.4). The research team can list the research questions based on the structure of the section 3.

3. I think the research team has already put a lot of effort to search and review the papers, and I do like to see more interesting findings from the outcomes. For example, the reasons for using travel time, distance or capacity from those previous articles. A lot of articles use the categories variable on geographic access, and just a few articles use the continuous variable. Do they have some reasons behind this?

4. Actually, the meta-analysis has some limitations. I suggest the authors can add more discussion on the limitation of method of this study.

6. PLOS authors have the option to publish the peer review history of their article (what does this mean?). If published, this will include your full peer review and any attached files.

Reviewer #1: No

Reviewer #2: **Yes: **Hsin Chung Liao

---

## [Author Response · Author response to Decision Letter 0]

22 Oct 2021

From:

Benoit Conti (corresponding author)

Laboratoire Ville, Mobilité, Transport 

Université Gustave Eiffel

benoit.conti@univ-eiffel.fr

Manuscript Number: PONE-D-21-10021 

To:

Editorial board,

Plos One

Paris, France, 22th October 2021

Dear Editor-in-Chief,

Dear Tzai-Hung Wen,

We would like to thank you for accepting the attached original revised manuscript entitled “Influence of geographic access and socioeconomic characteristics on breast cancer outcomes: a systematic review” for your consideration towards publication in Plos One.

In response to the previous review, we have carefully examined the comments made by the reviewers. All corrections and explanations required have been added to the revised manuscript. We feel that we have answered the reviewers’ questions, commented on the remarks point by point in the “responses to reviewers” document and modified our paper accordingly. All modifications appear in the manuscript highlighted in yellow and are included in the revised manuscript. Our article is a systematic literature review which does not contain any (original) datasets. In the Data Availability Statement, we reported that our article does not have data and the data availability policy is not applicable in our article. 

The word count for the main text is now 5,487 without references. The paper includes 3 tables and 7 figures in the main text. All correspondence may be directed to Benoit Conti at the address below. We would be delighted to provide any further information that may be required. We hope you will consider this paper suitable for publication in your journal. We look forward to receiving your editorial decision.

Yours sincerely,

Benoit Conti, on behalf of the authors

 

Manuscript Number: PONE-D-21-10021

REVISION

Influence of geographic access and socioeconomic characteristics on breast cancer outcomes: a systematic review 

We thank the editor and the reviewers for their positive comments on our paper. We have addressed the concerns that have been raised and we provide below a point-by-point reply to the comments of the two reviewers.

Reviewer #1

This study conducted a systematic review on the relationships between geographic access and socioeconomic status on breast cancer outcome. These results show that the better geographic access to healthcare facilities had a significant fewer mastectomy. But for the residential level SES, the deprivation index exhibited inconsistent relationship with stage at diagnosis. It is recommended to consider the suggestions below before probably publishing.

We thank the reviewer for his/her appreciation of our review paper and his/her suggestions for improvement. We have taken into account all the comments of the reviewer and provide below our point-by-point answers.

1. In Section 3.1, the authors could add a part to general discuss the information about socioeconomic characteristics.

Reply: As suggested by the reviewer, we have added information about demographic and socioeconomic characteristics. In the section 3.1, the following paragraph has been added: Socioeconomic characteristics were assessed at individual and residential levels: fifteen studies combine data from both levels, five papers present only individual level data and five papers use data at residential level only. At individual level, the most common characteristics used were age (n=20), ethnicity (n=13), partner status (n=8), health insurance (n=7) and education level (n=2). At residential level, the main data used were residential disadvantage or deprivation (n=10): six papers adopted existing deprivation indicators (e.g., Index of relative socioeconomic Advantage and Disadvantage [IRSAD], index of multiple deprivation [IMD]), and four papers created their own deprivation index (29, 31, 32, 45). Other variables used were poverty (n=5), educational level (n=4) and per capita income (n=2).

2. In Table 1, there is a typo in the column name "Relationship Odds ration [CI] or coefficients," and for the row Jones et al., 2008, the odds ratio did not fall into the range of 95% CI, for Onitilo et al., 2013, the upper value of 95% CI is lacking of a decimal point, these should be typos.

Reply: We thank the reviewer for his/her comment. The elements of the table have been modified.

3. According to Table 1, could the authors conclude the major confounding variables that must be considered for the future studies?

Reply: As suggested by the reviewer, we have added information about confounding variables for the future studies. In the results section, a new sentence now reads: Based on table 1, age and tumor features could be assessed as major confounding variables in the relationships between SES, geographic access and breast cancer outcomes. 

4. For Table 2, the references of travel distance could be sorted by initial letter to let the table more readable.

Reply: As suggested by the reviewer, the elements of the table have been modified. The articles have been arranged in ascending order by the variables. 

5. In Line 192-193, can the authors explain the reason why higher access to healthcare facilities have poorer survival rates?

Reply: As suggested by the reviewer, we have added the reasons given by the authors of the 2 studies mentioned. The following sentences have been added: the authors of the UK study put forward that this result may be “an artefact of imperfect control of the effects of deprivation, since inner city populations tend to be more deprived and closer to hospitals than suburban or rural populations” (26). In the state of Parana in Brazil, the authors suggested that the municipalities close to the services of specialized treatment in oncology were also areas with high population concentration which makes access to treatment difficult (28). 

6. In Line 245-247, how did the marriage status affect the odds of diagnosis stage and the odds of receiving BCS?

Reply: As suggested by the reviewer, we have added hypotheses about how marital status affects the odds of diagnosis stage and the odds of receiving BCS. We have also corrected errors in legend of the plots and in the manuscript. The paragraph in Section 3.3 now reads: The majority of the relationships between marital status and breast cancer outcomes were not significant (36, 38, 44). In two studies, married women or with partner had lower risks of late-stage diagnosis of breast cancer than other women (25, 37). In two papers, married women or with partner had higher odds of receiving BCS than single women (23, 44). According to the authors of these studies, these findings reflect the underlying issue of social support networks and social incentives which may affect women's motivation or ability to screen and/or to receive BCS.

7. In Line 281-283, could you explain why the less deprived neighborhoods would have greater odds of late-stage breast cancer diagnosis than more deprived neighborhoods?

Reply: We thank the reviewer for his/her comments about the article by Lian et al (2012). This paper provides an in-depth methodological analysis of the different methods of measuring geographic access (nine GIS-based measures). However, the authors do not offer an explanation of their findings on the link between deprivation and geographic access levels. We can hypothesise that socio-spatial distribution of different levels of deprived neighbourhoods in the St-Louis area could explain these results.

8. Authors could consider hospital classification as a surrogate to discuss the relationships of geographical accessibility to healthcare facilities.

Reply: The reviewer rightly points out to consider the hospital classification as a surrogate to discuss the relationships of geographical accessibility to healthcare facilities. In line with these comments, we have added:

• a paragraph in the first part of the discussion section. This new paragraph now reads: In addition, the level of specialization, the volume of surgery and/or the type of hospital (private/public) can influence surgical treatment. For instance, Dasgupta et al. (2016) showed that women who have significantly more likely to undergo breast reconstruction (following mastectomies) attended high-volume or private hospitals (and were younger, diagnosed more recently, had smaller tumors, lived in less disadvantaged or more accessible areas).

• some elements in the last sentence of the first part of the discussion section; now reads: To overcome this limit, it would be useful to include other places than home such as the workplace and any frequently visited places as well as to analyze women's health seeking behaviors including characteristics of the hospital used. 

9. For Conclusion section, can the authors give a recommendation on what kind of specific intervention or modification strategy for the stakeholders could take?

Reply: As suggested by the reviewer, we have added recommendations for stakeholders in the conclusion section. This new paragraph now reads: A major issue of future strategies should be to identify deprived patients at an early stage to implement corrective measures and care management adapted to each level of deprivation. These measures could be geographic (such as opening up in low medical density areas) and/or social (systematic referral of patients to social services of the hospital, work on perceptions of the disease and treatment) and/or medical (promoting participation in clinical trials, provide treatment side effects, facilitate access to supportive care).

10. Could the authors give a brief summary about which proxy is more suitable in accessing geographic accessibility and what is the limitation of this kind of theme and how can they improve in the future studies?

Reply: As suggested by the reviewer: 

• We have considered which proxy is the most suitable to evaluate geographic accessibility at the end of the systematic review. However, as the results of the 25 papers are very heterogeneous whatever the indicator used, it is difficult to define one as the most adequate. In section 3.2, we have added this point at the end of an existing sentence: Regardless of the type of measures used to calculate geographic access to health facilities (distance, time or geographical capacity), the heterogeneity of the results is very similar and does not allow to define which proxy is the most appropriate to assess geographic access.

• The use of proxy for assessing geographic accessibility has limitations since it reflects a theoretical accessibility from home to the closest facility. In order to get a more precise picture of women's care pathways, it is therefore necessary to take into account the facilities where women actually seek care. This point is developed in the subsection of the discussion “Taking account of the fact that people do not necessarily use the closest facility, and that they do not necessarily start from homeé that we have rephrased a bit. 

Instead of: “as stated by Khan-Gates (13), we need to further explore the “the actual geographic patterns of seeking care rather than access to the nearest facilities”. In addition, the actual geographical patterns of women need to include information about women’s lifestyle pathways, their reported travel time and their reasons for choosing (or not choosing) certain healthcare facilities.”

=> “To overcome this limitation of theoretical accessibility, future studies will ensure that women's actual care pathways are taken into account including information about their reported travel time and their reasons for choosing (or not choosing) certain types of healthcare facilities. As stated by Khan-Gates (13), we need to further explore the “the actual geographic patterns of seeking care rather than access to the nearest facilities”.

Reviewer #2: 

1. The authors use the meta-analysis method on 25 studies which focus on the Socio-economic and geographical inequalities in breast cancer mortality. Therefore, I like to review this study according to the purpose, contribution, and the limitation.

We thank the reviewer for his/her appreciation of our review paper and his/her suggestions for improvement. We have taken into account all the comments of the reviewer and provide below our point-by-point answers.

2. The main purpose of this study is to synthesize the current evidence of relationships between breast cancer outcomes and geographic access according to SES characteristics by using the meta-analysis method. I suggest the authors to list several specified research questions. Those specified questions can make your readers much more understanding the purpose of this study. Actually, this paper has already provided some basic findings on the general results (3.1), Relationship between geographic access and breast cancer-related outcomes (3.2), Relationship between SES characteristics and breast cancer-related outcomes (3.3), and Relationships according to geographic access measures, SES characteristics and breast cancer outcomes (3.4). The research team can list the research questions based on the structure of the section 3.

Reply: As recommended by the reviewer, we have modified each sub-title of section 3 to provide specific research questions based on the structure of the section and in line with our general research questions (Introduction section (L80 to 85)). The structure of section 3 now reads as follows: What measures of breast cancer outcomes, geographic access and SES characteristics? (3.1), What are the relationships between geographic access to health-care facilities and breast cancer outcomes? (3.2), What are the relationships between SES characteristics and breast cancer outcomes? (3.3), What are the combined effects of geographic access and SES characteristics on breast cancer outcomes? (3.4).

3. I think the research team has already put a lot of effort to search and review the papers, and I do like to see more interesting findings from the outcomes. For example, the reasons for using travel time, distance or capacity from those previous articles. A lot of articles use the categories variable on geographic access, and just a few articles use the continuous variable. Do they have some reasons behind this?

Reply: We thank the reviewer for his/her positive comments of our review paper. The reviewer asks for some clarification about the reasons given by the authors for using i) travel time, distance, or capacity and ii) continuous/categorical variables as measure of geographic access to healthcare facilities. In all articles that we have included in our review, the authors provide no reason about the statistical form (continuous/categorical) of the geographic access measures. The (potential) reason to explain using travel time and/or travel distance is methodological. Both measures were calculated using Geographical Information Systems (GIS software), but the estimation of travel time requires the input of a detailed road network dataset provided information such as road geometry, road restrictions and, travel speed (i.e., derived from road classification). Authors used capacity models (6 papers) to include population demand and healthcare provision in spatial models (e.g., spatial gravity-based model - as reported in Lines 172-173).

4. Actually, the meta-analysis has some limitations. I suggest the authors can add more discussion on the limitation of method of this study.

Reply: As suggested by the reviewer, we have added information about the limitations of the method. The paragraph now reads: Our systematic literature review has a number of limitations. First, using a quality assessment tool introduced some challenges. There is no consensus as to whether one should judge the representativeness of these characteristics of the study and the quality of the reviewed studies is based on what the authors reported in the paper. The quality assessment may not reflect a low quality of the study but might merely have been a lack of reported detail in the paper. Second, the heterogeneity of sample size, characteristics of the sample and measurement tools (both access and SES measures) limited the inter-study comparisons. Third, as in any systematic review, it is possible that some eligible studies may have been missed in our search strategy.

---

## [Decision Letter · Decision Letter 1]

31 Jan 2022

PONE-D-21-10021R1Influence of geographic access and socioeconomic characteristics on breast cancer outcomes: a systematic reviewPLOS ONE

Dear Dr. Conti,

Thank you for submitting your manuscript to PLOS ONE. After careful consideration, we feel that it has merit but does not fully meet PLOS ONE’s publication criteria as it currently stands. Therefore, we invite you to submit a revised version of the manuscript that addresses the points raised during the review process.

We look forward to receiving your revised manuscript.

Kind regards,

Tzai-Hung Wen, Ph.D.

Academic Editor

PLOS ONE

Journal Requirements:

Reviewers' comments:

Reviewer's Responses to Questions

6. Review Comments to the Author

Reviewer #1: Thanks for addressing my previous comments. The current version looks good to me. I don't have further questions.

Reviewer #2: 1. In the introduction part, the authors propose two questions in this study. First, in the context of equal geographic access to healthcare facilities, do women with disadvantaged social and economic characteristics have poorer breast cancer outcomes than more advantaged women? Second, in the case of equal socioeconomic level, do women with poor geographic access to healthcare facilities have worse breast cancer outcomes than women with higher geographic access? However, the authors present the results by 4 parts which included: What measures of breast cancer outcomes, geographic access, and SES characteristics? What are the relationships between geographic access to health-care facilities and breast cancer outcomes? What are the relationships between SES characteristics and breast cancer outcomes? What are the combined effects of geographic access and SES characteristics on breast cancer outcomes? I suggest the questions which are proposed in introduction should include those four questions to match the structure of results.

2. If the study brings the thought on taking account of the fact that people do not necessarily use the closest facility, and that they do not necessarily start from home. I suggest the authors can give some advice on the methods or measurements to the future studies. How should we take account those facts into our researches?

3. The same issue as the second comment. If women who live in suburban or rural areas are more willing to travel longer distances than women living in urban centers. How can we design our method to explore this possibility or realty? The authors can give the readers some advices on some arguments which are this study discussed.

4. The conclusion part has already mentioned some directions for the future studies. However, I strongly suggest the research team can give the readers all the implications which can response to all interesting results from your analysis.

---

## [Author Response · Author response to Decision Letter 1]

14 Feb 2022

From:

Benoit Conti (corresponding author)

Laboratoire Ville, Mobilité, Transport 

Université Gustave Eiffel

benoit.conti@univ-eiffel.fr

Manuscript Number: PONE-D-21-10021R1 

To:

Editorial board,

Plos One

Paris, France, 14th February 2022

Dear Editor-in-Chief,

Dear Tzai-Hung Wen,

We would like to thank you for accepting the attached original revised manuscript entitled “Influence of geographic access and socioeconomic characteristics on breast cancer outcomes: a systematic review” for your consideration towards publication in Plos One.

In response to the previous review, we have carefully examined the comments made by the second reviewer. All corrections and explanations required have been added to the revised manuscript. We feel that we have answered the reviewer’s questions, commented on the remarks point by point in the “responses to reviewers” document and modified our paper accordingly. All modifications appear in the manuscript highlighted in yellow and are included in the revised manuscript. 

The word count for the main text is now 5,672 without references. The paper includes 3 tables and 7 figures in the main text. All correspondence may be directed to Benoit Conti at the address below. We would be delighted to provide any further information that may be required. We hope you will consider this paper suitable for publication in your journal. We look forward to receiving your editorial decision.

Yours sincerely,

Benoit Conti, on behalf of the authors

 

Manuscript Number: PONE-D-21-10021

REVISION

Influence of geographic access and socioeconomic characteristics on breast cancer outcomes: a systematic review 

We thank the editor and the reviewers for their positive comments on our paper. We have addressed the concerns that have been raised and we provide below a point-by-point reply to the comments of the two reviewers.

Reviewer #1

Thanks for addressing my previous comments. The current version looks good to me. I don't have further questions.

Reply: We thank the reviewer for his/her appreciation of our review paper and his/her previous suggestions for improvement.

Reviewer #2: 

We thank the reviewer for his/her appreciation of our review paper and his/her suggestions for improvement. We have taken into account all the comments of the reviewer and provide below our point-by-point answers.

1. In the introduction part, the authors propose two questions in this study. First, in the context of equal geographic access to healthcare facilities, do women with disadvantaged social and economic characteristics have poorer breast cancer outcomes than more advantaged women? Second, in the case of equal socioeconomic level, do women with poor geographic access to healthcare facilities have worse breast cancer outcomes than women with higher geographic access? However, the authors present the results by 4 parts which included: What measures of breast cancer outcomes, geographic access, and SES characteristics? What are the relationships between geographic access to health-care facilities and breast cancer outcomes? What are the relationships between SES characteristics and breast cancer outcomes? What are the combined effects of geographic access and SES characteristics on breast cancer outcomes? I suggest the questions which are proposed in introduction should include those four questions to match the structure of results.

Reply: As recommended by the reviewer, we have modified the questions which are proposed in introduction and included the four questions to match the structure of results (Introduction section (L85 to 90)). The end of the introduction now reads as follows: To answer these two general questions, the result section will be divided into four research questions: (i) what measures of breast cancer outcomes, geographic access, and SES characteristics? (ii) What are the relationships between geographic access to health-care facilities and breast cancer outcomes? (iii) What are the relationships between SES characteristics and breast cancer outcomes? (iv) What are the combined effects of geographic access and SES characteristics on breast cancer outcomes?

2. If the study brings the thought on taking account of the fact that people do not necessarily use the closest facility, and that they do not necessarily start from home. I suggest the authors can give some advice on the methods or measurements to the future studies. How should we take account those facts into our researches?

Reply: We thank the reviewer for his/her comments. A sentence already present in the article offers an initial response to your comment in the discussion section (L360-364). As recommended by the reviewer we add a sentence on this methodological issue (L364-366): To this purpose, information on where women come from when they go to hospital and why they choose (or not choose) their hospital should be included both in questionnaire and interview.

3. The same issue as the second comment. If women who live in suburban or rural areas are more willing to travel longer distances than women living in urban centers. How can we design our method to explore this possibility or realty? The authors can give the readers some advices on some arguments which are this study discussed.

Reply: We thank the reviewer for his/her comments. As noted by the reviewer, considering the local residential context is needed to better understand travel distance to healthcare. One option could be to systematically characterize this local context with information from self-reported women (questionnaire) or/and from spatial databases on urban density (e.g. urban, suburban, rural). In this way, we add a sentence on this methodological issue (L409-411): To this purpose, information on urban density level of residential area of women should be collected either from self-reported questionnaires or from spatial databases (e.g., urban, suburban, rural areas).

4. The conclusion part has already mentioned some directions for the future studies. However, I strongly suggest the research team can give the readers all the implications which can response to all interesting results from your analysis.

Reply: We thank the reviewer for his/her comments. As recommended by the reviewer we add a sentence about the implications in the conclusion section: This requires a specific research protocol based on regular questionnaires and/or interviews at different times, from diagnosis to one or more years later, including dimensions of precariousness and ability to travel as well as spatial access to healthcare (e.g. address of their general practitioners (GP), possession of a driving licence, availability of public transport).

---

## [Editor Report · Decision Letter 2]

29 Jun 2022

Influence of geographic access and socioeconomic characteristics on breast cancer outcomes: a systematic review

PONE-D-21-10021R2

Dear Dr. Conti,

We’re pleased to inform you that your manuscript has been judged scientifically suitable for publication and will be formally accepted for publication once it meets all outstanding technical requirements.

Kind regards,

Tzai-Hung Wen, Ph.D.

Academic Editor

PLOS ONE
---

## [Editor Report · Acceptance letter]

8 Jul 2022

PONE-D-21-10021R2 

Influence of geographic access and socioeconomic characteristics on breast cancer outcomes: a systematic review 

Dear Dr. Conti:

I'm pleased to inform you that your manuscript has been deemed suitable for publication in PLOS ONE. Congratulations! Your manuscript is now with our production department. 

Kind regards, 

on behalf of

Dr. Tzai-Hung Wen 

Academic Editor

PLOS ONE